# Natural Gas Liberations around Production Wells at Russian Arctic Gas Fields

## Vladimir S. Yakushev 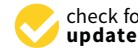

Gas Fields Development Department, Gubkin University of Oil and Gas, 119991 Moscow, Russia;
yakushev.v@gubkin.ru

**Abstract:** Gas samples from gas liberations around wellheads on two giant natural gas fields in West Siberia (Bovanenkovo and Yamburg) have been tested for their carbon isotopic and molecular compositions. Results have shown local, microbial genesis of gas and that its source is permafrost at both gas fields. Gas liberation is caused by permafrost rock massif thawing around working well. Gas liberations can appear at different distances from the casing inside the radius of thawing. Two gas samples taken from gas liberations at casing border have shown thermogenic origin, which was explained by deep gas leakage through the casing. Gas liberations from deep production horizons are few, and they concentrate around the casing. Permafrost gas liberations are numerous, and they are spread at different distances from the wellhead.

**Keywords:** permafrost gas; methane carbon isotopic composition; gas molecular composition; gas production well; thawing radius

## 1. Introduction

Development of oil and gas fields in permafrost regions of Russia has led to the discovery of an unexpected phenomenon: gas releases of different intensities from the permafrost zone during well drilling and operation. Permafrost was supposed to be impermeable to natural gas due to ice filling the pore space [1]. Oil and gas operators believed that these liberations were caused by gas leaks through the production column (casing). This means that a well is not completed properly, and it should be closed. Information about these gas liberations was very poor and unofficial. Nevertheless, some data could be found in well-drilling journals and in records of well operator observations [2]. Collection of these data from West Siberia giant gas fields had continued for a long time. Some regularities of gas releases have been established: each gas field at the North of West Siberia has production wells with gas liberations, and the number of such wells is increasing toward the North. Gas releases can increase when a well starts production and decrease if the well is stopped [2]. To detect the source of this gas, molecular and isotopic composition analysis was required. It is known, that the majority of shallow subsurface gas accumulations in a permafrost area is represented by microbial methane [2–6]. Methane is a dominant component in permafrost gas (up to 99%). More heavy hydrocarbons are not represented practically, unlike in subpermafrost production horizons. So gas composition can indicate a source of gas liberation around a wellhead.

## 2. Observations and Methods of Study

Methane has no smell and is not visible. Visually detecting gas liberations around a wellhead can only be done in the spring time, because spring water accumulates in ground depressions around the working well and gas bubble chains become visible (Figure 1). Unfortunately, gas liberations around working wells are only visible a few weeks at the end of May and beginning of June. To collect samples around wells in other seasons is not possible because no visible indicators are available: in winter,

snow covers ground depressions, and in summer, the depression is dry. But year-by-year observations of the same liberations in springtime prove their constant activity all over the year if a well continues to work and the permafrost massif around continues to thaw. Chains of bubbles have constant character when a well is under operation and have different intensities. Permafrost thawing around working wells results in gas releases appearance far from the wellhead (up to 5 m, Figure 2). Sometimes, when a well was out of work for a while, gas liberations decreased, indicating a direct link between rock thawing and gas release.

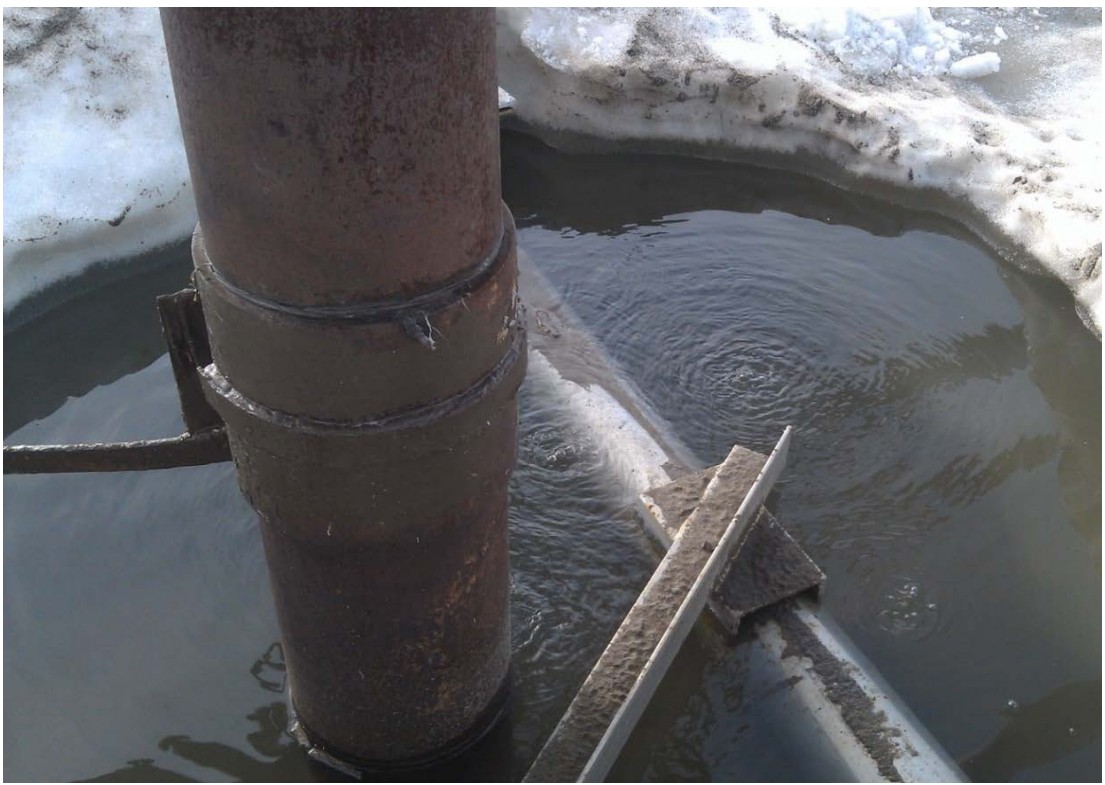

**Figure 1.** Gas releases in spring water around working gas well [7].

Due to the great amount of gas, passing through water, the gas sampling procedure was simple: a plastic bottle (200–250 cm$^3$), filled with water, submerged into spring water near a wellhead, and gas bubbles penetrated inside the open neck of the bottle. When the bottle was filled half by gas, it was closed under the water surface by a screw-on lid. Gas samples were stored and transported in a reverse position (bottleneck down).

Gas samples were collected in two giant Arctic gas fields: Yamburg (six samples) and Bovanenkovo (nine samples) (Figure 3). These fields have similar typical production well construction, but different times of operation: Yamburg has been under operation since 1985, and Bovanenkovo since 2013. The thawing radius around each production well is different. Also, the upper production horizon is located at depth 1200 m at Yamburg and 600 m at Bovanenkovo, so gas has to pass different distances to the surface if leaking through the casing. Permafrost thickness at Yamburg is 450 m, which in combination with geothermal conditions, provides sufficient pressure for the presence of a methane Hydrate Stability Zone (HSZ). At the Bovanenkovo field, the thickness of the permafrost layer is about 200 m, and it is not enough for the existence of a HSZ. The average annual permafrost temperature at Yamburg is −5 °C, and at Bovanenkovo it is −7 °C. Gas composition in the upper production horizon (Cenomanian) is similar in both fields: 98–99% methane and some admixture (less than 1%) of $N_2$ and $CO_2$ [8]. Isotopic composition of methane carbon $\delta^{13}C$ in the upper productive horizon is also similar for both fields: −44 to −55% [9].

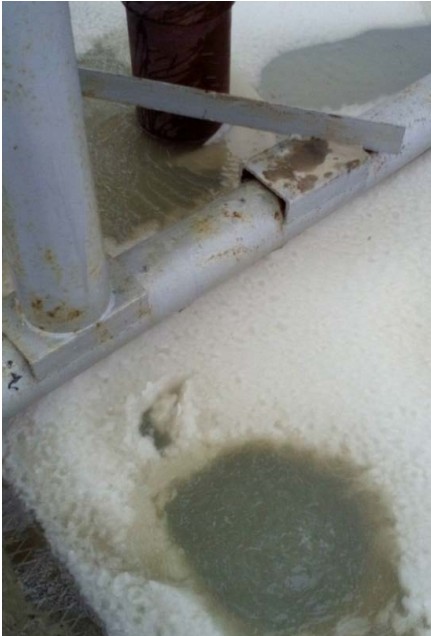

**Figure 2.** Distant gas release from thawed permafrost massif around working gas well at the Yamburg gas field [10].

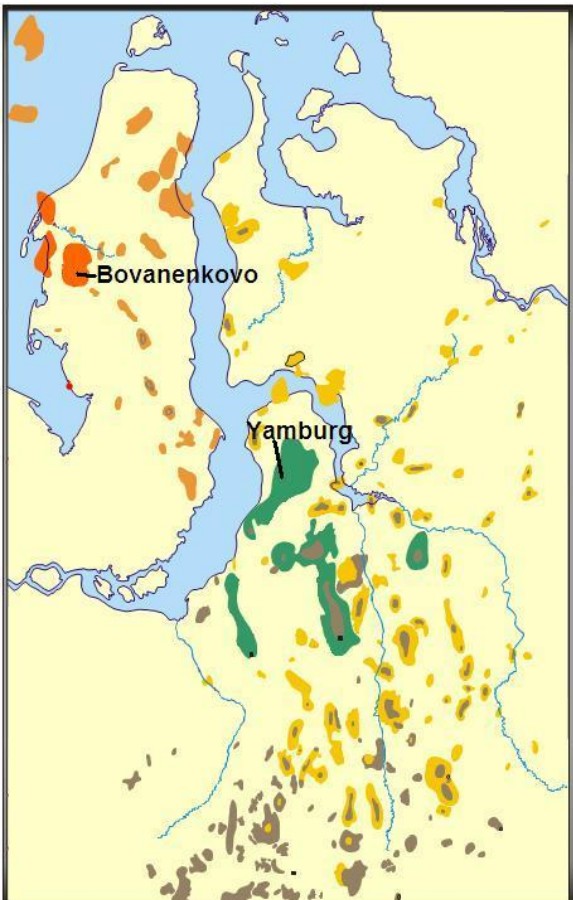

**Figure 3.** Locations of permafrost gas sampling around gas wells in West Siberia (Different colors indicate groups of gas fields).

Isotopic composition of carbon in gas samples from around each well was measured by different equipment. The first measurements for gas from the Bovanenkovo field in the 1990s were made with a MI-1305 mass-spectrometer (USSR). Molecular analysis of Bovanenkovo gas was performed by a Tsvet 100 chromatograph (USSR). The general procedure of measurements for such studies is described in [9]. Gas was sampled from a permafrost section when well drilling [2].

Samples from the Yamburg gas field were tested by a Delta V Advantage mass-spectrometer (Finnigan, Germany). Chromatographic analysis was performed with a Trace GC Ultra (Thermo Fisher Inc., Milan, Italy) gas chromatograph, equipped with a Isolink console with an oxidizing reactor. Component separation was performed in a CP-PoraPLOT column at 100 °C with a constant flow of helium. Then the gas sample was fed to an oxidizing reactor, where $CO_2$ was produced from hydrocarbon components (if any). Carbon dioxide was fed to the mass-spectrometer through a ConFlo IV interface. That gas was sampled around gas wells [10] that were working over a long time (more than 30 years).

## 3. Results and Discussion

The following data on molecular composition were received for the Bovanenkovo and Yamburg permafrost gas samples (Table 1).

**Table 1.** Molecular (component) composition of permafrost gas samples (% vol., ranges).

| Component | Yamburg | Bovanenkovo |
|:---:|:---:|:---:|
| $CH_4$ | 91–99 | 97–99 |
| $N_2$ | 1–9 | 1–3 |
| $CO_2$ | traces | traces |
| Other ($H_2$, $H_2S$, heavy hydrocarbons) | traces | traces |

The molecular composition of permafrost gas samples is different from Cenomanian gas due to high $N_2$ content, but also it is characterized by the presence of microbial gas [11]. The source of the gas in the permafrost section has to be investigated by examining the isotopic ratio of carbon in the methane. Few gas samples from working wells were taken to compare the compositions from wellhead ground depressions and from the casing inside the wells. The isotopic composition of methane carbon is presented in Table 2.

**Table 2.** Methane carbon isotopic ratio ($\delta^{13}$C, %, ranges) in gas samples.

| Place of Sampling | Yamburg | Bovanenkovo |
|:---:|:---:|:---:|
| Outside Casing (Permafrost Gas) | −70 to −71 | −70 to −74 |
| From Inside casing | −50 to −51 | No data, but according to [9] −44 to −55 |

In two cases, gas samples were taken outside, but very close to the gas condensate well casing at the Yamburg field, and have shown an isotopic composition corresponding to the gas-condensate production horizon [9]. This means that the wells had some leaks through casing [10].

It is obvious that the isotopic composition of gas inside a well and outside a well is different if no leak occurs through the casing. Both composition data indicated a microbial genesis of gas from the permafrost section around production wells. This means that the gas inside permafrost is separated from deeper production horizons with thermogenic gas. However, powerful natural gas blowouts from depths of 30–50 m have been discovered in Yamal peninsula close to Bovanenkovo field [12] bring up the question: how can considerable volumes of gas be generated and concentrated in the environment, which is unfavorable for microbial activity (inside permafrost)? Two possible answers are: microbial gas is concentrated in permeable rock layers during the perennial freezing of geologic section or gas penetrated to permafrost depth interval from deeper horizons, changing its isotopic and

molecular compositions on its way. The second answer does not have much supporting evidence: only large volumes of methane are concentrated in the permafrost section of the Yamal peninsula, where deep natural gas fields are discovered below the permafrost. These methane accumulations could be the result of gas migration from deeper horizons. But permafrost gas composition measurements made in areas with no deep layers of thermogenic gas, have shown considerable microbial gas content in frozen sediments without any connection with thermogenic gas [5,6,13]. Microbial gas generation has been proved by numerous studies (see, for example, [14–16]). Therefore, a local microbial origin of permafrost gas is the most probable. This gas can be concentrated during the perennial freezing of geologic sections or taliks inside the permafrost [2,17]. But the majority of this microbial gas is remains disseminated in the permafrost section in free and hydrate [18] forms, filling the pore space of permafrost rocks [2]. Heat flow from working production wells causes thawing of pore ice and hydrates in these rocks releasing the included gas [19]. Gas finds the most permeable channels in thawed rocks around a well column and moves upward, forming gas "seeps" around the wellhead when it is under operation.

## 4. Conclusions

The isotopic and molecular composition of gas samples around production wells in the permafrost area of West Siberia indicates local, microbial genesis of this gas. Gas "seeps" in ground depressions around working wells are a result of the heat interaction of warm wells and the surrounding permafrost. The heat from wells causes a thawing of the surrounding rock massif. Gas encapsulated earlier in pore space in free or hydrate forms by ice is then released and rises up toward the ground surface, forming permeable channels in the thawed rock massif. Long-time operation of a well increases the thawing radius and causes new releases of earlier trapped gas. This gas can find an exit to surface far from the well casing at the border of the thawed zone, unlike thermogenic gas from the deep production horizon (if leaking through the casing), which releases along the outer wall of the casing.

**Funding:** The research was supported by Gubkin University special funding.

**Conflicts of Interest:** The author declares no conflict of interest.

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
