# Peer review of "Natural Gas Liberations around Production Wells at Russian Arctic Gas Fields"

_geosciences, doi:10.3390/geosciences10050184_

Round 1
Reviewer 1 Report
This study used the molecular and stable carbon isotopic composition of gases released via bubbling next to gas infrastructure in northwestern Siberia. The study concludes that the methane released is predominantly of biogenic (microbial) origin from permafrost layers, rather than thermogenic origin from deeper natural gas production layers.
The study's analysis approach and methodology is, in principle, sound. And the conclusions are supported by the strong isotopic evidence of a microbial methane signature. However, the manuscript is extremely short and I think inadequately describes relevant research in the introduction as well as the current methodology.
There appears to be quite a comprehensive literature regarding methane seeps in Siberian gas wells. More of this should be used to expand the introduction. In particular, greater reference should be made to the mechanisms of permafrost methane persistence/stability. The broad applicability of this research to the question of the global methane budget and its uncertainties should also be described. For example, of what potential significance are these methane fluxes?
In terms of methodology and results, the author must state more about the timing of the sample collection and more information about the methods used to analyze the samples. What detectors are used, and what reference gases or calibrations were used? The data reported in Table 2 are convincing, but are the numbers ranges or confidence intervals around mean values? I am also concerned that the number of samples analyzed may be too few, or that they are not distributed in a logical time. Is it possible that the contribution of different methane sources could change seasonally? These questions are difficult to address without more information about the timing of sample collection - though presumably it happened during summer.
Overall, this is a promising study. Uncertainties in the global methane budget are a timely topic. The problem is that the scholarship and reporting is too cursory, and possibly the dataset is insufficient for a publication venue of this stature.
Author Response
Please, find attached file. My answers are colored by yellow.

Reviewer 2 Report
The reviewed paper has very significant value both for fundamental and applied studies in the Arctic regions. Author identified the process of disturbance around the production wells as a potentially very important mechanism of release greenhouse gases trapped in permafrost to the atmosphere. But, at the same time, the importance of this question requires a more detailed discussion and deeper analysis of results.
Here are some suggestions which might help the author to improve the paper:
General - I would recommend replacing the words "permafrost rocks" to just "permafrost".
Include in the "Introduction" section a brief description of the natural conditions of the study area. Especially some data about the spatial variability of tpermafrost temperature must be mentioned as an important factor that affects the dimensions of the disturbance zone around the wells.Â
In the "Observations and methods of study" section author tells only about sampling procedure around the wells, but later (line 72) author tells that permafrost also was sampled for gas. This moment requires some explanations.
Lines 55-56: What are the typical dimensions of the "thawing radius around production wells"?
Lines 58-59: I would recommend changing the sentence to "Permafrost thickness at Yamburg is 450 m which, in combination with geothermal conditions provides sufficient pressure for the presence of methane Hydrate Stability Zone (HSZ). At the Bovanenkovo thickness of the permafrost layer is about 200 m and it is not enough for existence of HSZ. "Â
Lines 71-75. Could you, please, give some brief equipment specification (accuracy, resolution etc)?
Tables 1 and 2: Give more statistics information (mean values, standard error and deviation).
Line 96: I recommend replacing the word "medium" with "environment".
In the section "Results and Discussion" author mentions two possible mechanisms of the biogenic gas accumulation in permafrost: during the freezing of deposits and in-situ formation in the frozen ground. The first one is supported by the references, but the second was not. I would recommend to author to check the following publications (
Heslop, J. K., Matthias Winkel, K. M. Walter Anthony, R. G. M. Spencer, D. C. Podgorski, P. Zito, A. Kholodov, M. Zhang, and Susanne Liebner. "Increasing organic carbon biolability with depth in yedoma permafrost: ramifications for future climate change." Journal of Geophysical Research: Biogeosciences 124, no. 7 (2019): 2021-2038. and
Rivkina, E. M., Friedmann, E. I., McKay, C. P., & Gilichinsky, D. A. (2000). Metabolic activity of permafrost bacteria below the freezing point. Applied and Environmental Microbiology, 66(8), 3230–3233.), which confirm the possibility of methanogen metabolic activity under the negative temperature.
Also would comparison of the results of the presented research with other studies performed in Siberian Arctic (
Kraev, Gleb, Elizaveta Rivkina, Tatiana Vishnivetskaya, Andrei Belonosov, Jacobus van Huissteden, Alexander Kholodov, Alexander Smirnov, Anton Kudryavtsev, Kanayim Teshebaeva, and Dmitrii Zamolodchikov. "Methane in gas shows from boreholes in epigenetic permafrost of Siberian Arctic." Geosciences 9, no. 2 (2019): 67.) will improve the paper a lot.
In conclusion, I would recommend accepting the paper for publication with minor revision.
Author Response

(The authors gave the same response as above.)

Round 2
Reviewer 1 Report
Thank you for explaining the difference between your studies and the previous work. The additions to the methods are also helpful.